# Cyclodextrin–Drug Inclusion Complexes: In Vivo and In Vitro Approaches

**DOI:** 10.3390/ijms20030642

**Published:** 2019-02-02

**Authors:** Simone Braga Carneiro, Fernanda Ílary Costa Duarte, Luana Heimfarth, Jullyana de Souza Siqueira Quintans, Lucindo José Quintans-Júnior, Valdir Florêncio da Veiga Júnior, Ádley Antonini Neves de Lima

**Affiliations:** 1Chemistry Department, Amazonas Federal University, Av. Rodrigo Octavio, 6200, Manaus AM 69080-900, Brazil; Braga.simone.c@gmail.com; 2Department of Pharmacy, Federal University of Rio Grande do Norte, Natal RN 59012-570, Brazil; fernandailary@gmail.com; 3Laboratory of Neuroscience and Pharmacological Assays (LANEF), Federal University of Sergipe, São Cristóvão SE 49100-000, Brazil; luahei@yahoo.com.br (L.H.); jullyanaquintans@gmail.com (J.d.S.S.Q.); lucindojr@gmail.com (L.J.Q.-J.); 4Military Institute of Engineering, Praça General Tiburcio, 80, Praia Vermelha, Rio de Janeiro RJ 22290-290, Brazil; valdir.veiga@gmail.com

**Keywords:** cyclodextrins, substances, inclusion complexes, pharmaceutical technology, biological assays, drug delivery

## Abstract

This review aims to provide a critical review of the biological performance of natural and synthetic substances complexed with cyclodextrins, highlighting: (i) inclusion complexes with cyclodextrins and their biological studies in vitro and in vivo; (ii) Evaluation and comparison of the bioactive efficacy of complexed and non-complexed substances; (iii) Chemical and biological performance tests of inclusion complexes, aimed at the development of new pharmaceutical products. Based on the evidence presented in the review, it is clear that cyclodextrins play a vital role in the development of inclusion complexes which promote improvements in the chemical and biological properties of the complexed active principles, as well as providing improved solubility and aqueous stability. Although the literature shows the importance of their ability to help produce innovative biotechnological substances, we still need more studies to develop and expand their therapeutic properties. It is, therefore, very important to gather together evidence of the effectiveness of inclusion complexes with cyclodextrins in order to facilitate a better understanding of research on this topic and encourage further studies.

## 1. Introduction

In 1891, Antoine Villiers first isolated oligosaccharides produced by starch or starch derivatives using the enzyme cyclodextrin (CD) glycosyltransferase. Cyclodextrins (CDs) have, therefore, been known for more than 120 years, but only really took off in the 1980s with the first applications in the pharmaceutical and food industries. This expansion occurred with industrial scale production of the three cyclodextrins: alpha-cyclodextrin (α-CD), beta-cyclodextrin (β-CD), and gamma-cyclodextrin (γ-CD). They were produced in pure form as early as 1984, lowering prices, which contributed to their development, especially that of β-CD [1,2]. CDs used to form host–guest inclusion complexes with various drugs in solution or a solid state have been recognized as pharmaceutical excipients [3,4,5].

In respect of their chemical structure, cyclodextrins have a cavity size which is determined by the number of glucose units. The free hydroxyls on the outside of the CDs impart a more hydrophilic character, whereas the oxygen atoms in the glycosidic bonds and the hydrogen atoms impart a hydrophobic character within the cavity, which allows the dissolution in aqueous medium of compounds with low solubility. In addition, the space inside the molecule of the cyclodextrins allows the formation of inclusion complexes with compounds that present little solubility [2,3,5].

The natural cyclodextrins α-, β-, and γ-CDs are composed of 6, 7, or 8 glucose units (Figure 1A), and their synthetic derivatives are divided into three groups: hydrophilic, such as 2-hydroxypropyl-β-CD (HP-β-CD); hydrophobic, such as 2,6-di-O-ethyl- β-CD; and ionizable, such as sulfobutylether β-CD (SBE-β-CD) [1,5,6].

CDs are widely used in pharmaceuticals, drug delivery systems, cosmetics, and the food and chemical industries. They can be found in commercially available medications, including tablets, eye drops, and ointments [3,4,7,8,9,10,11,12]. Their successful use in inclusion complexes with bioactive compounds has led to extensive investigations in several different application areas to try to overcome the limitations of certain substances (Figure 1B) [1,2,5,13,14].

The use of CDs in the food industry is well established, being used in the reduction of cholesterol in food, notably in dairy products; as dietary fibers, useful for controlling body weight and blood lipid profile, and as prebiotics, which enhance the intestinal microflora by selective proliferation of bifidobacteria [15,16]. Although CDs are either non or only partly digestible by the enzymes of the human gastrointestinal (GI) tract and fermented by the gut microflora, they produce negligible cytotoxicity (mainly β-CD) in foods. CDs have also been investigated for use as smart active packaging of foods [16,17,18].

Despite being considered GRAS (Generally Recognized as Safe) substances, CD safety and toxicity usually depends on the route of administration and the type of CD used. When given orally, CDs are negligibly absorbed from the GI tract, thus are practically nontoxic due their bulky and hydrophilic nature; however, higher doses of CDs may be harmful and produce irreversible kidney damage and dysfunction [19,20].

Moreover, CDs like HP-β-CD and SBE-β-CD are considered safe for parenteral administration and are commonly used with antitumor and immunomodulatory drugs [21,22,23]. Recently the Food and Drug Administration (FDA) approved pharmaceutical formulations containing liposomal preparations of doxorubicin (Doxil), daunorubicin (DaunoXome), cytarabine (DepoCyt), and amphotericin B (Abelcet), which have proven to be attractive and less toxic alternatives to the conventional drug formulations [19]. Therefore, CDs are well tolerated, atoxic if used in safe concentration ranges already well-described in the literature and considered safe for complexion with drugs.

Inclusion complexes formed with a host–guest molecule may exhibit improved chemical or biological properties compared to the host molecule alone. Such inclusion may: (i) improve aqueous solubility, dissolution, and bioavailability [24]; (ii) increase the physicochemical stability of drugs and improve the shelf life of drugs [3,24]; (iii) modify the drug delivery site and/or the time profile [3,4,5]; (iv) reduce or eliminate unpleasant taste and smell [3,4,5]; (v) prevent drug–drug or drug–excipient interactions [4]; and (vi) convert liquid drugs into microcrystalline or amorphous powders [13].

In relation to the biological activities of compounds complexed with CDs, it is essential to evaluate the effect of the drug in the complexed and non-complexed states, using an equivalent dose of the drug or equivalent weight [25,26]. In vitro and in vivo studies of inclusion complexes with CDs may provide evidence of their therapeutic effect. This review highlights studies using in vitro assays to evaluate the inclusion complexes with CDs in relation to their antimicrobial, antichagasic, antitumor, and antioxidant properties, and in vivo bioassays examining their anti-inflammatory, antinociceptive, anticancer, intestinal absorption, and other characteristics.

The literature reports several examples of natural and synthetic compounds complexed with CDs, showing their ability to produce innovative products for the food and pharmaceutical industries; however, due to the need for a better understanding of the therapeutic properties of inclusion complexes with CDs, it is essential to bring together studies which focus on their biological and therapeutic effectiveness. This review aims to highlight and provide a critical review of the biological performance of natural and synthetic substances complexed with CDs.

## 2. In Vivo Cyclodextrin Studies

This literature search was performed using specialized databases (ScienceDirect and SciFinder) using different combinations of the following keywords: “Cyclodextrins”, “Cyclodextrins in vivo” and “Cyclodextrins in vitro”. The inclusion criteria used were: both in vitro and in vivo studies that investigated biological action and the use of CDs to improve the biological performance of complexed substances. Figure 2 shows the in vivo biological activities of the substances complexed with CDs. 

### 2.1. Anti-Inflammatory Activities of Cyclodextrin Inclusion Complexes

There are several diseases that activate the inflammatory process, and consequently a growing interest in substances with anti-inflammatory properties, in particular cyclodextrin inclusion complexes (CD-IC) which have used innovative biotechnology to increase the bioavailability, solubility and pharmacological effect of drugs [27,28,29]. The CD most commonly found in studies of CD-IC with anti-inflammatory activity is β-CD because of its good capacity for complexation with anti-inflammatory drugs, its appreciable oral acceptability, and low cost [27,30]. Recently, some interesting systematic reviews have strongly suggested that β-CD is a tool that can increase the bioavailability of these drugs and also produce better efficacy for analgesic and anti-inflammatory drugs [24,27]. 

In an attempt to treat inflammatory disorders, anti-inflammatory compounds have been used; these drugs are typically classified as non-steroidal anti-inflammatory drugs (NSAIDs) and corticosteroids. Many of these drugs have serious side effects, such as gastrointestinal disturbances, neutropenia and cardiovascular risks [31]. CDs can enhance the effect of anti-inflammatory drugs while also improving side effect profiles [27]. Table 1 shows the in vivo activities of CD-IC in the studies in this review.

A number of studies report the anti-inflammatory efficacy of CD-IC containing anti-inflammatory compounds. A study was carried out with the incorporation in a CD-IC of *Copaífera multijuga* oleoresin (CMO), comprising mainly β-caryophyllene (β CP), which has anti-inflammatory properties. CMO was incorporated into β-CD and hydroxypropyl cyclodextrin (HP-β-CD) by the kneading (KND) and slurry (SL) methods. Powder X-ray diffraction, Fourier transform infrared spectroscopy (FTIR), thermal analysis, and scanning electron microscopy were used to evaluate the interactions of the CMO with the CDs. Physicochemical characterization confirmed the formation of inclusion complexes of CMO with β-CD and HP-β-CD by the KND and SL methods. To evaluate the anti-inflammatory efficacy of the inclusion complexes, carrageenan-induced paw edema in mice was used to evaluate the CMO alone and complexed with CDs as shown in Figure 3. With the anti-inflammatory activity in vivo, it was possible to observe that the CMO assay outside and within an inclusion complex, at a concentration of 100 mg/kg maintained its anti-inflammatory activity, with a reduction of paw edema, nitric oxide and myeloperoxidase, without alterations in the anti-inflammatory activity of CMO [32].

The pentacyclic triterpenoid pedunculoside (PE), is known for its anti-inflammatory properties. However, its potential is limited due to low oral bioavailability. To reduce this problem and evaluate anti-inflammatory activities, it was incorporated in a CD-IC. Anti-inflammatory studies were performed with ear edema of mice induced by dimethyl-benzene and Croton’s oil. The PE and the PE–CD-IC exhibited anti-inflammatory effects induced in mouse ear edema induced by dimethyl benzene and croton oil. Furthermore, treatment with 90 mg/kg of PE–CD-IC improved the degree of ear edema more effectively than 20 mg/kg of PE in the murine models of inflammation. It was found that PE in CD-IC has low toxicity and higher aqueous solubility, contributing to substance absorption in vivo, and was, therefore, more effective in mouse ear edema [52].

Studies with the essential oil of the *Ocimum basilicum* (OBEO) plant incorporated in CD-IC demonstrated anti-inflammatory and anti-edematogenic effects. Anti-inflammatory efficacy was observed with oral administration of the OBEO complexation with β-CD. To evaluate the anti-inflammatory activity of OBEO/β-CD the following models were used: paw edema induced by different agents and evaluated by plethysmometry; vascular permeability assessed by Evans Blue die; peritonitis induced by carrageenan and a chronic inflammation model of granuloma formation induced by cotton pellets. The doses of OEOB/β-CD were chosen based on results obtained with pure OBEO, using an uncomplexed dose of 50 mg/kg. The complex presented significant inflammatory effects at 10 mg/kg. At this dose, the effects obtained with orally administered OBEO/β-CD were similar to those obtained with the treatment with pure OBEO at 50 mg/kg. As a result of this conjugation, the OBEO presented anti-edematogenic and anti-inflammatory activities that were similar to those observed in mice treated with pure OBEO at a 5 times higher dose. The results of the study indicated that this complex could be used in anti-inflammatory drug development [28].

In order to increase aqueous solubility, kamebakaurin (KA) was complexed with β-CDs. The characterization techniques ultraviolet/visible spectroscopy, nuclear magnetic resonance (NMR), infrared spectroscopy (IR), X-ray diffraction (XRD) and Scanning Electron Microscopy (SEM) were used to evaluate the behavior of the CD-ICs and the interaction of KA with β-CDs. The water solubility of KA was improved in the presence of β-CDs, and the characterization studies showed the positive development of the KA inclusion complex with β-CDs. Results of in vivo studies for anti-inflammatory activity of KA/β-CDs have shown their potential for use in anti-inflammatory applications [29].

A study was done into the influence of natural β-CD and its hydrophilic derivatives (HP-β-CD and SBE-β-CD) on the in vitro dissolution rate, in vivo absorption rate, and oral bioavailability of the water-soluble anti-inflammatory agent valdecoxib (VALD). Equimolar drug/CD solid complexes were prepared by the kneading and coevaporation methods. To examine whether the notable increase in dissolution rate observed with CD based formulations may lead to differences in pharmacological effects, the anti-inflammatory profiles of VALD, VALD–β-CD and VALD–SBE-β-CD complexes were explored using the carrageenan-induced rat hind paw edema model. CD complexes (equivalent to 1 mg VALD kg) or the drug alone (1 mg/kg) were administered orally as aqueous suspensions with 0.25% carboxymethyl cellulose. VALD alone showed a slow in vivo absorption rate with a maximum inhibition of edema (16%) after a period of 3 h. In contrast, VALD included in the cavity of both the CDs showed a higher absorption rate in vivo, achieving a more than 50% inhibition of edema in 1h and a maximum inhibition (66%) after a period of 3 h. These pharmacological evaluations in rats indicated that VALD–CD complexes might be used to develop a new solid oral formulation with an in vivo performance much better than that of VALD alone [33].

Another study developed inclusion complexes of meloxicam with β-CD- and β-CD-based nanosponges to enhance their solubility and stability, and to prolong release using different methods (physical mixing, kneading and sonication). Their anti-inflammatory activities were assessed using the carrageenan-induced paw edema model (1% *w/v* carrageenan suspension). One group received meloxicam orally at a dose of 1 mg/kg and another group received a dose of 3 mg/kg. The meloxicam alone reduced inflammation in the model. The anti-inflammatory activity of the complexes was much greater than the drug alone and suggested that controlled release of meloxicam from the nanosponges occurred [34].

A study with coumestrol (COU) was conducted to evaluate its effectiveness in wound healing. The study was performed with the complexation of COU with HP-β-CD to improve the bioavailability and solubility of the COU, as it has low aqueous solubility, which is a crucial limitation for biological tests. Fibroblasts and an in vitro experimental artificial wound model were used to compare the effects of the free COU and the complex with HP-β-CD. The 50 μM (66.1%)/10 μM (56.3%) COU/HP-β-CD complex induced proliferation and cell migration in the wounds. The in vivo experimental model of healing (Wistar rats) revealed that COU/HP-β-CD incorporated into a hypromellose hydrogel (HPMC) showed similar efficacy in wound healing compared to the positive control (Dersani^®^), with a healing benefit of 50% being achieved within a shorter time period [35].

### 2.2. Antinociceptive Activities of Cyclodextrin Inclusion Complexes

The search for products with improved antinociceptive properties has intensified, with the insertion of active principles into substances such as CDs to form IC in order to develop more effective products with better stability, solubility, and bioavailability, and a consequent improvement in their pharmacological properties [25,36,37,38,39,40,41,53,54].

(−)-Linalool (LIN) is a monoterpene derived from plants, mainly from the Lauraceae and Lamiaceae families, and has anti-inflammatory and analgesic activities. As it is a terpene, it presents low solubility and chemical stability, reducing its clinical application. To improve its aqueous stability and solubility and pharmacological effects, LIN was complexed with β-CD. The antinociceptive effects of LIN and LIN/β-CD in concentrations (40 mg/kg), were evaluated using the acetic acid, formalin and hot plate test. Strong antinociceptive activity was demonstrated in all the tests (*p* < 0.01 or < 0.001). The antinociceptive activity of LIN/β-CD was higher when compared to LIN alone [53].

*Lippia grata* essential oil (EO) was complexed with β-CD/essential oil (β-CD/EO) to test its efficacy in the treatment of orofacial pain. The characterizations by Differential scanning calorimetry (DSC) and Thermogravimetry (TG) showed the complexation of the *L. grata* essential oil in β-CD. The antinociceptive effect was evaluated in orofacial pain models in mice, with concentrations β-CD/EO of 6, 12, or 24 mg/kg. The results demonstrated that treatment with β-CD/EO was capable of reducing nociceptive face-rubbing behavior in both phases of the formalin test. This result may be related to the terpenes in the oil such as camphor, borneol and β-caryophyllene. The antinociceptive activity may have occurred through activation of the motor cortex, the nucleus raphe pallidus (NRP) and the periaqueductal gray (PAG) (brain areas involved in pain modulation) [37].

The antihyperalgesic activity of (−)-linalool alone and complexed in β-CD (LIN-CD) was evaluated in an animal model of fibromyalgia (FM) consisting of chronic non-inflammatory muscular pain and its effect on the central and nervous system. The anti-hyperalgesic effect induced by LIN-CD (25 mg/kg) lasted for 24 h, unlike the effect of treatment with LIN alone (25 mg/kg). The central nervous system (CNS) areas involved in the anti-hyperalgesic activity were evaluated by immunofluorescence. LIN or LIN-CD produced a significant reduction in mechanical hyperalgesia in a chronic, non-inflammatory muscle pain model. These compounds significantly (*p* < 0.05) activated neurons of the locus coeruleus, nucleus raphe magnus, and periaqueductal gray areas. The development of new therapeutic options for the treatment of chronic pain, mainly those related to the modulation of neurotransmission downstream of pain control, may be viable given the positive results of LIN-CD in the management of pain in FM [38].

A public health problem is cancer pain, which itself affects the quality of life of patients as, in addition, do the reported side effects of existing therapeutic options. Carvacrol (CARV), is a monoterpene that has been studied for the control of painful conditions and inflammation. The use of this terpene can be improved with β-CD incorporation. The effect of CARV complexed in β-CD on nociception was evaluated in rodents with tumor cells (Sarcoma 180). The antihyperalgesic and nociceptive effects of the complex were evaluated in mice with a tumor in the paw. CARV/β-CD complex was administered (50 mg/kg) in mice with a tumor in the hind paw and was shown to be able to reduce hyperalgesia for 24 h, unlike the free CARV (100 mg/kg), which promoted effects which only lasted 9 h. Administration on alternate days of the CARV/β-CD complex (12.5–50 mg/kg) reduced hyperalgesia, as well as spontaneous and palpation-induced nociception. However, pure CARV (50 mg/kg) did not cause significant changes in nociceptive responses. The results showed that the encapsulation of CARV in β-CD may be useful for the development of new options for the treatment of pain [39].

The antihyperalgesic effect of a complex containing β-caryophyllene and β-CD (βCP–βCD) was evaluated in chronic muscular pain in rodents. The characterization of the complex was performed by DSC, TG, FTIR, XRD, and SEM, and these analyses showed that the incorporation of β-caryophyllene into β-CD was efficient. When tested in chronic non-inflammatory muscular pain in mice, the complex promoted the inhibition of Fos protein in the lumbar spinal cord at concentrations of (10 and 20 mg/kg). Oral treatment with βCP–βCD, at all doses tested, produced a significant (*p* < 0.05) reduction in mechanical hyperalgesia and a significant (*p* < 0.05) increase in muscle withdrawal thresholds, without producing any alteration in force. In addition, βCP–βCD was able to significantly (*p* < 0.05) decrease Fos expression in the superficial dorsal horn. The results suggest that cannabinoids have the ability to reduce neuronal excitability in the sensory neurons that cause pain and thus contribute to the antihyperalgesic effect [25].

Hecogenin acetate (HA) has been shown to have a good analgesic profile, but low water solubility reduces its use in chronic conditions. To test whether the complexation of hecogenin into β-CD (HA-CD) improves the chemical and pharmacological properties of this apolar compound, HA was complexed with β-CD and characterized by thermal, morphological and spectroscopic analysis. Oral HA or HA-CD treatment produced significant antinociceptive profiles and also decreased mechanical hyperalgesia at concentrations of (20 mg/kg). Treatment with HA or HA-CD produced a significant antinociceptive (*p* < 0.01) profile and also decreased mechanical hyperalgesia, with HA-β-CD showing significantly better effects when compared to HA alone (*p* < 0.05). The data corroborate the idea that CDs may be an important tool in improving the analgesic profile of non-polar compounds [40].

Additionally, the antihyperalgesic profile of D-limonene, a terpene found in citrus fruits, was optimized after β-CD complexation and produced an effect lasting more than 8 h, 2 h longer than the non-complexed D-limonene [42] (Figure 4). The immunofluorescence for Fos protein showed that treatment with D-limonene significantly decreased the number of Fos-positive cells in the dorsal horn of the spinal cord when compared to the control group, suggesting this antihyperalgesic effect is mediated by the involvement of descending pain-inhibitory mechanisms (Figure 4C–F). These effects on the descending pain pathways appear to be common in terpenes complexed with CD and improve analgesic action and pharmacological efficacy when compared to non-complexed terpenes [15,36,53]. Interestingly, using a docking approach it was demonstrated that the possible forms of this terpene are accommodated in the complexed the β-CD [55]. This was confirmed by Nuclear Magnetic Resonance (NMR), so this property is commonly attributed to compounds having increased bioavailability by CDs.

One study reported the development of inclusion complexes of meloxicam with β-CD and β-CD based nanosponges to try to increase their solubility and stability, and prolong drug release. Particle size, zeta potential, encapsulation efficiency, stability studies, in vitro and in vivo drug release studies, FTIR, DSC, and XRPD were used to characterize parameters. Swiss albino mice (20–25 g) of both sexes were used to test the analgesic activities of the formulation and meloxicam. The analgesic activity in the mice was determined by counting the number of writhes induced by 0.6% acetic acid (10 mL/kg, i.p.). A greater percentage of inhibition (71.11 ± 1.47%) of the writhing responses was observed with the complex than with meloxicam alone (45.55 ± 2.05%). This was reported to be due to increased inhibition of cyclooxygenase and inflammatory mediators by the complex compared to meloxicam alone [34].

### 2.3. Anticancer Activities of Cyclodextrin Inclusion Complexes

To increase bioavailability and solubility, betulinic acid, a very promising anti-melanoma agent, was complexed with water-soluble γ-CD. The physical chemical characterization was done by DSC, X-ray and SEM. Its antiproliferative activity was assessed by 3-(4,5-dimethylthiazol-2-yl)-2,5-diphenyltetrazolium bromide (MTT) assay and cell cycle analysis. An animal model of murine melanoma in mice was used to test the complex (100 mg/kg, intraperitoneally). The results showed a reduction in tumor volume and weight, showing that this complexation was beneficial in antiproliferative activity and the reduction of tumor development in vivo [55].

Curcumin (CUR) has anti-carcinogenic potential and has been complexed with β-CD in order to improve its bioavailability for cancer chemoprevention. In vivo studies were performed using rats injected with lung cancer cells, with a curcumin dose of 100 mg/kg. The curcumin-β-CD complex enhanced curcumin delivery and improved therapeutic efficacy compared to free curcumin in vivo and in vitro [43].

Phenoxodiol was complexed with β-CD to overcome its poor water solubility, which limits its efficacy as an anticancer agent. The complex was investigated against three different cancer cell lines, all neuroblastoma cells. The cells were treated 24 h after sowing at a concentration of 1 to 500 μM of the complex. After 72 h of drug incubation, the treatment medium was replaced with 10% Alamar blue in fresh medium and the cells were incubated for a further 6 h. The metabolic activity was detected by spectrophotometric analysis evaluating the absorbance of the Alamar blue. The aqueous solubility of the phenoxodiol in β-CD was improved and the in vitro biological evaluation revealed increased antiproliferative activity against the three cancer cell lines. In addition, the toxicity of the complex against normal human cell line was 2.5-fold lower. These data indicate that the encapsulation of phenoxodiol in β-CD leads to an improvement in its water solubility and biological activity [56].

Albendazole (ABZ) is an anticancer drug, but has poor aqueous solubility. ABZ was complexed with SBE-β-CD and HP-β-CD. In vivo anticancer studies were done with mice injected intraperitoneally with ovcar-3 cells. Two weeks later, treatment started with 50 mg/kg of ABZ-SBE-β-CD injected in a phosphate buffered saline (200 μL volume of solution) in group 1. The controls received only SBE-β-CD in group 2. Abdominal circumference remained almost constant up to 32 days with treatment starting at day 14. The volume of ascites in ABZ-SBE-β-CD treated group was significantly reduced (>50%) compared to the control (*p* = 0.0135). Thus, only ABZ-SBE-β-CD was effective in the control of ascites volume in this study. The tumor weight was measured at the end of the 36-day treatment period, the weight in the ABZ-SBE-β-CD group was 30% higher compared to the SBE-β-CD control, but was not statistically significant (*p* = 0.3012) [57].

A study with a γ-CD inclusion complex with γ-CD was performed with containing Caffeic acid phenethyl ester (CAPE), which is an anticancer bioactive component of propolis, was performed. The in vivo study was done with BALB/c nude mice (4 weeks old, female). Human cancer cells were injected subcutaneously into the abdomen of the nude mice. CAPE (200 mg/kg body weight) or propolis (250 mg/kg body weight) were as then injected orally. Tumor formation and body weight of the mice were monitored every other day. It has been reported that CAPE-γCD presented higher cytotoxicity for a wide range of cancer cells, is stable in an acid medium and is, therefore, recommended as a potent anticancer treatment [58].

### 2.4. Intestinal Absorption of Cyclodextrin Inclusion Complexes

CD complexation could be the solution for most active substances presenting low oral bioavailability. A study evaluated the intestinal permeability of tanshinone IIA (TSIIA), a substance with cardiovascular benefits, complexed with HP-β-CD. The complexes were obtained by coevaporation. Free and complexed TSIIA absorption was assessed using the everted intestinal sac technique in rats. The TSIIA was complexed at three concentrations (50, 100, 150 µg·mL^−1^) and administered in three intestinal segments (the duodenum, jejunum, and ileum). The absorption parameters of TSIIA at each concentration in the three intestinal segments were tested. It was observed that increasing the dosage of the complex caused saturation, suggesting that the transport mechanism in vivo is related to passive transport. Greater permeability was found for the complexed TSIIA in the intestinal membrane with increased oral bioavailability [51].

Genipin is an aglycone extracted from the fruit of *Gardenia jasminoides* Ellis, and has traditionally been used in China for the treatment of disease, and in industry it is used to produce dyes. With respect to its pharmacological effects, it has anti-inflammatory and antithrombotic properties, and has recently gained prominence for its possible antidepressant effect. However, low intestinal absorption has been a barrier to its therapeutic application. Researchers set out to investigate the effect on intestinal absorption of an inclusion complex of genipin with HP-β-CD. The complex was produced and physicochemically characterized by evaluating its in vitro dissolution profile. Intestinal absorption was evaluated by the direct perfusion method in rats, with administration of the equivalent of 50 mg/kg of free and complexed genipin. The study found that the dissolution properties of the complex were superior to the isolated drug, and the rates of intestinal absorption and permeability were considerably greater for the complex [58]. Due their bulky and hydrophilic nature, a negligible amount of intact CDs are absorbed from the GI tract by passive diffusion [59].

The question of how bile salt in the small intestine interacts with CDs has been investigated by researchers. An experiment was conducted using isothermal titration calorimetry to determine how various β-CD and bile salt interact in supramicellar concentrations (SMC). Analysis of the results showed that the direct interactions between the bile salt micelles and CDs were insignificant. From this knowledge, an extended form of Stella’s CDs utility number (UCD), was suggested to achieve total drug solubilization in the intestine where bile salts are present [60]. In another study, the pharmacokinetics of two complexed substances, danazol—which has high affinity for HP-β-CD and cinnarizine—which has low to medium affinity, were measured in vivo. The pharmacokinetic study was performed by intravenous and oral administration of the CD-IC, accompanied by increasing administration of uncomplexed CD. Cinnarizine did not undergo significant changes in pharmacokinetic parameters. The study concluded that the risk of CD overdose is relatively low in vivo when demonstrating substances with stability constants in a normal range [61].

## 3. Studies In Vitro with Cyclodextrin Inclusion Complexes

As already described in this article, the lack of water solubility reduces the flexibility for drug formulation and administration, thus CDs haves been used successfully to improve these properties of drugs with poor solubility. The hemolytic effect of CDs has been reported in several in vitro studies; but the toxicological implication in vivo is considered small [61]. Therefore, it is clear that in vivo and in vitro approaches should take into account essential aspects such as the action of the uncomplexed free drug alone, whether the complexed CD-drug acts directly, and whether the CD-IC is dissociated in the in vitro environment, among other aspects. Table 2 shows the in vitro activities performed with the CD-IC in the studies in this review.

Non-polar drugs are usually more soluble in in vitro assays after complexation with CDs than non-polar drugs alone, however, some compounds strongly bound to the CD, limiting their availability in in vitro assays and consequently the biological effects [62]. CDs can act as “secondary antioxidants,” improving the ability of traditional antioxidants to prevent enzymatic browning when assessed using in vitro assays [63]. The studies suggest that CD-IC are not usually degraded in in vitro tests, therefore, the biological properties are slightly different (mitigated or potentiated) compared to in vivo studies [64]. Figure 5 shows the in vitro biological activities of the substance complexes with CDs.

### 3.1. Antimicrobial Activities of Cyclodextrin Inclusion Complexes

Carvacrol, eugenol, linalool, and 2-pentanoylfuran were complexed with CDs and their solubility and antimicrobial activity against *Staphylococcus aureus*, *Bacillus subtilis*, *Escherichia coli*, and *Saccharomyces cerevisiae* were evaluated. The CDs increased the solubility of the compounds in an aqueous environment and increased their antimicrobial activity. Carvacrol was found to be the most effective antimicrobial compound with the lowest minimum inhibitory concentration—MIC (0.5 μg/mL). Eugenol and linalool displayed a high effect against Gram-positive bacteria, with an MIC of 2.5 μg/mL against *S. aureus*, and 1.25 μg/mL and 2.5 μg/mL against *B. subtilis*, respectively. The least effective was 2-pentanoylfuran but it did show the most promising MIC towards the yeast, *S. cerevisiae* (MIC 1.25 μg/mL). The addition of CDs had a marked effect on the antimicrobial activity of the EO compounds. The complexation altered the lipid composition of the main membrane of all the strains analyzed. Scanning electron microscopy revealed that cell integrity was significantly affected, resulting in cell lysis [78].

The essential oil of *Hyptis martiusii* (EOHM) is important therapeutically because it has antibacterial activity, but its low solubility and bioavailability compromise its use. β-CD was used as a pharmaceutical excipient to improve the physicochemical properties and low solubility of this oil. Antimicrobial assays with the strains *Staphylococcus aureus*, *Pseudomonas aeruginosa*, and *Escherichia coli* were performed using the microdilution method with 96-well microtiter plates. The MICs for the β-CD alone or associated with the EOHM demonstrated a MIC of 1024 mg/mL, except when used against *S. aureus* (MIC = 32 mg/mL). The essential oil alone demonstrated an antibacterial and modulating effect against *S. aureus*. However, the inclusion complex provided greater solubility of the oil, but did not demonstrate greater biological activity in the antibacterial assays performed [28].

A study aimed to characterize the carvacrol/β-CD inclusion complex, establish its physicochemical properties and evaluate its antimicrobial and antioxidant activities, since CDs can improve the solubility and dissolution of this water insoluble constituent by inclusion in the CDs’ hydrophobic cavities. CD-IC of carvacrol were prepared by the lyophilization and kneading methods, and were analyzed using DSC and phase solubility analysis. Both methods showed high entrapment efficiencies, drug loading, and increased solubility in water. Minimum inhibitory concentrations (MICs) for the inclusion complexes and pure carvacrol were determined using a broth dilution assay using *Escherichia coli* and *Salmonella enterica* cultures. The concentration of the carvacrol in the inclusion complexes used in the tests ranged from 2000 to 18,000 μg/mL (200 to 1800 μg·g/mL of carvacrol concentration based on the entrapment efficiency), while the concentrations of free carvacrol ranged from 250 to 1600 μg/mL. Minimum inhibitory concentration and minimum bactericidal concentration (MIC and MBC) values for free carvacrol were 1150 and 1000 μg/mL for *S. typhimurium* and *E. coli*, respectively. MIC values for HP-β-CD encapsulated carvacrol showed improvement in inhibition (MIC) ranging from 60 to 74% (*p* < 0.05) more than free carvacrol for both pathogens. Moreover, bactericidal activity was also improved (*p* < 0.05) for both pathogens, thereby reducing the carvacrol concentration needed to be effective. Both types of complexes inhibited *E. coli* and *S. typhimurium* at a lower concentration than free carvacrol, indicating that encapsulation may enhance the mechanism of antimicrobial action and decrease the concentration of antimicrobial compound required for inhibition [79].

Another study presented the preparation, characterization and properties of a chrysin-β-CD inclusion complex. The stoichiometry of the inclusion complex was established to be 1:3 (chrysin to β-CD), with the inclusion rate of 90.5 ± 2.63 at 55 °C. Antibacterial activity was analyzed by the inhibition halo test, measuring the diameter of the zone of inhibition. Chrysin and chrysin-β-CD had a limited inhibitory effect on food spoilage bacteria such as *E. coli*, *B. subtilis*, *S. aureus* and *Salmonella*, with the antibacterial effect of chrysin-β-CD being weaker than that of chrysin. The diameter of antibacterial zone of chrysin was 6.18 mm, 6.12 mm, 6.09 mm, and 6.11 mm, respectively, for the four bacteria. While the diameter of antibacterial zone of chrysin-β-CD was 6.05 mm, 6.02 mm, 6 mm, and 6.03 mm, respectively [80].

### 3.2. Anticancer Activities of Cyclodextrin Inclusion Complexes

Aphidicolin (APH), a tetracyclic diterpene, exhibits specific cytotoxic action against neuroblast cells (NB). This diterpene is important for the development of new antitumor drugs because treatment failure in most patients with NB is related to primary or acquired resistance to conventional chemotherapeutic agents. One study compared the antitumor efficacy of APH in parental cell lines and cell subclones that showed drug resistance to vincristine (VCR), doxorubicin (DOX), and cisplatin. To improve the solubility of APH in water, gamma-cyclodextrin APH IC (gamma-CD) was used for the systemic treatment of xenotransplanted parental tumors and resistant VCRs. The cells used in the study were neuroblast cells. The inclusion complex with a concentration of 2 mg/mL of APH presented antitumoral action about 2-fold lower than that of the free drug. APH and its IC-gamma CDs reduced the growth of NB cells in vitro. The growth of tumors sensitive to NB and VCR was inhibited in equal doses in a dose-dependent manner in vivo [23].

A curcumin–cyclodextrin complex improved curcumin delivery and enhanced its therapeutic efficacy compared with free curcumin in in vitro assays with lung cancer cells, which seemed to involve, at least in part, the mitogen-activated protein kinase (MAPK)/NF-κB pathway associated with increased Bax/caspase 3 expression. Cell viability was determined by a colorimetric assay using MTT 3-(4,5-Dimethyl-thiagol-2yl)-2,5-diplenyltertrazollium. The MTT assay showed that free curcumin and the β-cyclodextrin-curcumin complex inhibited human lung cancer cell growth. Moreover, the 50% inhibitory concentration (IC_50_) of free curcumin was higher than that of the β-cyclodextrin-curcumin 24 h and 48 h after incubation [56].

A study presented the preparation, characterization and properties of a chrysin-β-cyclodextrin inclusion complex. The stoichiometry of the inclusion complex was established to be 1:3 (chrysin to β-cyclodextrin), with an inclusion rate of 90.5 ± 2.63 at 55 °C. The MTT method was used to explore the inhibition of mouse hepatoma H22 cells by the chrysin–β-cyclodextrin inclusion complex. The ability of chrysin and the inclusion complex to inhibit H22 tumor cells had no obvious dependency on concentration. When the chrysin sample concentration was 0.25 mol/mL, the inhibition rate was 33.52%. The inclusion complex reached a maximum inhibition rate of 25.64% at 1 mol/ mL [80].

In another study, the bioavailability by oral administration of fisetin (FST), a potent anticancer phytoconstituent, was enhanced by encapsulation into PLGA NPs (poly-lactide-co-glycolic acid nanoparticles) and association into inclusion complexes containing HP-β-CD. This increased the peak plasma concentration and total drug absorbed, improving its anti-cancer activity against breast cancer cells [81].

In vitro cytotoxicity of ABZ-SBE-β-cyclodextrin in ovarian tumor cell line OVCAR-3 and human ovarian epithelial cells Ovcar-3 cells was evaluated. The absorption of 8.0 mg/mL solution containing 7.0 mg/mL (ABZ) at 40% CD at 25 °C, showed the maximum complexity with SBE-β-CD and HP-β-CD, mainly at three days, with stability of 2 weeks. [ABZ] complexed with SBE-β-CD exhibited potent cytotoxicity (in vitro and in vivo) in ovarian tumor cells [57].

### 3.3. Antichagasic Activity of Cyclodextrin Inclusion Complexes

Benznidazole (BNZ) has low solubility and high toxicity but is a very promising drug for the treatment of Chagas disease, and it is essential to find technological solutions for these problems of solubility and toxicity. Therefore, ICs were developed and characterized in binary systems (BS) with BNZ and randomly methylated cyclodextrin (RMCD), and in ternary systems (TS) with BNZ, RMCD, and hydrophilic polymers. The results showed that solid BS showed a large increase in dissolution rate (Q > 80%). The CD-IC obtained by the kneading method at a ratio of 1:0.17 (77.8% over 60 min) appeared to be the most suitable method for the development of a solid oral pharmaceutical. The use of cyclodextrins was shown to be a viable tool for effective, standardized and safe drug delivery [65]. Solid dispersion (SD) studies were performed with hydroxypropyl methylcellulose (HPMC) and β-CD in order to increase the solubility/dissolution of BNZ in water. The BNZ was incorporated with CDs using the physical mixture (PM), kneading (KND), evaporation, and atomization methods. The analyses were based on techniques of in vitro dissolution and molecular modeling. Molecular modeling showed that BNZ could form β-CD complexes in different ways, such as in aqueous solution or in a vacuum. In vitro dissolution showed improved BNZ solubility in inclusion complexes produced by the PM, SD, and KND methods with HPMC, and also that β-CD IC produced by the SD method had improved solubility. The β-CD IC is more effective in promoting the improvement of the solubility of BNZ compared to the SD with HPMC and may increase the bioavailability of the drug and improve its pharmaceutical potential [82].

One study analyzed the trypanocidal activity of BNZ alone and complexed with methylated-β-cyclodextrin (RM-β-CD). The IC_50_ values obtained for free BNZ and for BNZ: RM-β-CD were 0.037 ± 0.2 mM and 0.027 ± 0.4 mM, respectively. The results showed that the growth inhibition performances of BNZ and RM-β-CD were similar and dose and time dependent, achieving a 92% inhibition of growth for both BNZ and 0.027 mM complexes. The BNZ complexation with CDs maintained the trypanocidal activity, which is of great importance. The BNZ inclusion complex with CDs is a promising alternative for the development of a new, safe and stable product for the treatment of Chagas disease [83].

Currently only two drugs are available for treatment: nifurtimox and benznidazole. Nifurtimox is no longer prescribed in many countries, such as Brazil, Chile, Uruguay, and USA, owing to reports of gastrointestinal effects, neurotoxicity, genotoxicity, and low efficacy against some *T. cruzi* strains. Benznidazole, 2-nitro-*N*-(phenylmetil)-1H-imidazole-1-acetamide, proved efficient in the initial period of the illness, but its efficacy in the chronic phase is controversial [84].

Previous studies have reported that nitroaromatic compounds have activity against *T. cruzi*. For this reason, the idea that an analogue of metronidazole1-(2-iodo ethyl)-2-methyl-5-nitroimidazole (MTZ-I) might also exhibit trypanocidal activity makes sense. However, it is not suitable for use in pharmaceutical formulations because of its low solubility. Thus, an inclusion complex with β-CD was produced using the physical mixture method with 1:1 stoichiometry. The samples were physicochemically characterized by phase solubility, DSC, FTIR and molecular modeling. The results showed an interaction between the MTZ-I and the β-CD, demonstrated by physicochemical modifications of the inclusion complex compared with MTZ-I alone. In vitro trypanocidal activity against amastigote and trypomastigote forms of *T. cruzi* were evaluated and a cytotoxicity study was performed. The results showed that MTZ-I: β-CD was 10 times more active than MTZ-I alone, demonstrating that the presence of an iodine atom on the side chain increased trypanocidal activity while leaving cytotoxicity unaltered [85].

### 3.4. Antioxidant Activity of Cyclodextrin Inclusion Complexes

Antioxidants present limited applicability due to their instability in light and oxygen, as well as their low solubility in water. Complexation with cyclodextrins can provide greater stability to these molecules which have great therapeutic potential. Studies have been done with catechin, whose potential applications are limited by its instability. One study produced inclusion complexes with β-CD, HP-β-CD, and CD-Mβ for application in food and health supplements. The complex with β-CD was used for the evaluation of stability, and physicochemical and antioxidant properties. The complexation with β-CD improved the solubility of catechin, and its stability against temperature, light, and oxygen. The antioxidant activity was measured by 2,2-diphenyl-1-picrylhydrazyl (DPPH) reduction. When DPPH accepts an electron donated by the DPPH antioxidant compound it is discolored, which can be measured quantitatively from the changes in absorbance. The inclusion complex was calculated as having an antioxidant activity of 24 mg, while that of catechin alone was 4.7 mg. The complex presented activity approximately three-fold greater than the activity of the pure compound [86].

Diosmin (DIOS) is a flavonoid that has strong antioxidant activity, as well as other important effects in inflammatory conditions, cancer and the treatment of ulcers. Its solubility limits its therapeutic applications. Thus, there is a need to develop a water-soluble formulation that allows better dissolution and absorption, and consequently greater bioavailability. This may also result in the development of pharmaceutical products with lower doses, because the dissolution efficiency leads to a higher percentage of the drug being absorbed. Complexes were obtained by the kneading method, with a stoichiometry of 1:1. The formation of the complex was confirmed by FTIR spectral analysis, XRD, DSC, SEM, and 1:00 NMR. The antioxidant activity was measured by DPPH, with the results indicating a significantly higher DPPH reduction rate when the DIOS was complexed with β-CD compared to pure DIOS. This enhanced antioxidant activity of the DIOS complex was probably due to higher solubility of the active binary complex, resulting in greater solubility, dissolution, and antioxidant activity [87].

In another study, the authors assessed the capacity of the extract obtained from the seeds of the milk thistle to penetrate the skin and its antioxidant activity after complexation in HP-β-CD (phytocomplex) and incorporation in appropriate formulations. The extract from MT consists mostly of a mixture of flavonoids called silymarin which have high antioxidant power in skin cells. Two water-in-oil emulsions, one enriched with dry milk thistle matter and melon (MT) and the other a binary complex composed of MT and HP-β-CD (1:4 p/p) were produced. They were used to determine the distribution on the skin and the influence of HP-β-CD dextrin on antioxidant activity and extract permeation. The antioxidant activity of the extract of MT was tested by DPPH reduction, showing a change in the activity of the extract after complexation and after incorporation in emulsions. It was observed that the binary complex decreased antioxidant activity of the phytocomplex when compared to the dry extract. This phenomenon occurs due to complexation of silymarin in the HP-β-CD cavity. The complex formation decreased the ability of flavonoids to react with DPPH within 60 min, and it is likely that beyond this period of time there was activity. Ex vivo and in vitro permeation tests showed a lower and more controlled rate of release of the phytocomplex when compared with the pure extract. The authors concluded that cyclodextrins appear to be good excipients for the complexation and silymarin, because there was less degradation of the flavonoids [88].

Ferulic acid (FA) is a highly effective antioxidant and photoprotector, already recognized as a sunscreen, but not suitable for use in cosmetic formulations due to its low physical and chemical stability. For ingredients used in cosmetics, the formation of inclusion complexes can increase stability, solubility, and delivery on the skin. The preparation of inclusion complexes of α-CD with FA by the co-precipitation method was realized from an aqueous solution with a stoichiometry of 1:1 and incorporation of this complex into an oil/water emulsion (o/w). The complex was analyzed by High performance liquid chromatography (HPLC), X-ray diffraction, ^1^H NMR, photostability, release, and antioxidant activity. The antioxidant activity of the complex was also assessed using ORAC (oxygen radical absorbance capacity), which measures the antioxidant inhibiting oxidation induced by peroxyl radical with a fluorescent probe, with the antioxidant capacity of a substance being determined by the rate of reduction of fluorescence over time. The results of this analysis showed that the antioxidant inclusion complex with FA had lower antioxidant power when compared to the FA. The authors correlated this fact firmly with the encapsulation of the FA inside the cavity of the α-cyclodextrin, making it less available for interaction with the peroxyl radical. There was a delay in the release of the FA complexed in emulsion. The authors claimed these results suggested that complexes topically applied as sunscreen would be safe and ensure long lasting protection from sun exposure [89].

Curcumin is a polyphenol with antioxidant, anti-inflammatory, and anticancer properties. However, its application is limited by its low solubility and oral bioavailability. Inclusion complexes using sulfobutylether-β-cyclodextrin (SBE-β-CD) were made. The complexes were obtained by different methods and their physicochemical parameters were assessed before an in vitro test in a cell line of human hepatic cancer (HepG-2) was carried out. In these cells, the antioxidant activity of the complex was assessed by measurement of cellular reactivate oxygen species (ROS), using 2,7′-dichlorofluorescin diacetate (DCFDA), a fluorogenic dye. Doxorubicin was used as a control ROS inducing substance. The cells were treated with SBE-β-CD by means of equivalent doses of curcumin alone, which ranged from 0.1–50 µM. The experiment showed an improvement in the antioxidant activity of CR as a consequence of complexation with SBE-β-CD, thanks to the considerable increase in its solubility [90].

## 4. Conclusions

Based on this literature review, it is clear that inclusion complexes with CDs have great potential in the pharmaceutical area, adding to the bioavailability, solubility, and stability of drugs, and improving their desired pharmacological effects and biological activities in vivo and in vitro; however, few studies have explored these improvements through pharmacokinetic studies and the use of molecular approaches. Despite this, the review emphasizes the importance of CD-IC because they represent potential biotechnological innovations, as well as providing evidence to support the importance of further studies to enable the enlargement of the therapeutic arsenal and greater pharmacological efficacy of existing treatments.

## Figures and Tables

**Figure 1 ijms-20-00642-f001:**
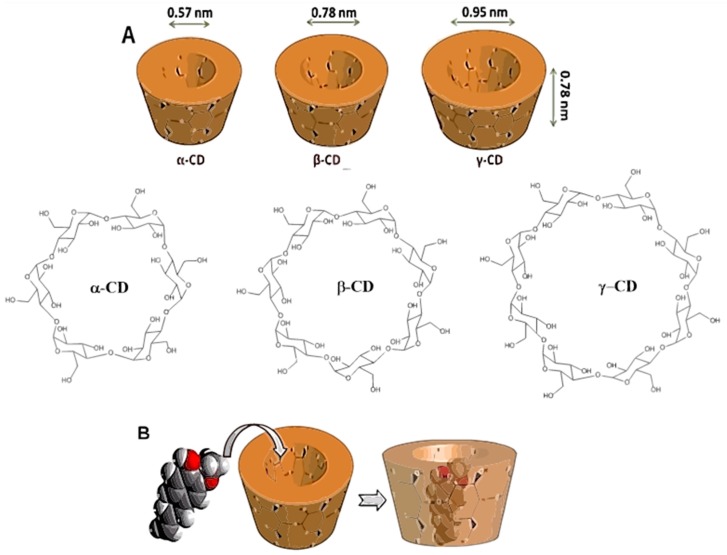
(**A**) α-, β-, and γ-CD (cyclodextrin) molecules; (**B**) Representation of compound inclusion complex in cyclodextrin [5,14].

**Figure 2 ijms-20-00642-f002:**
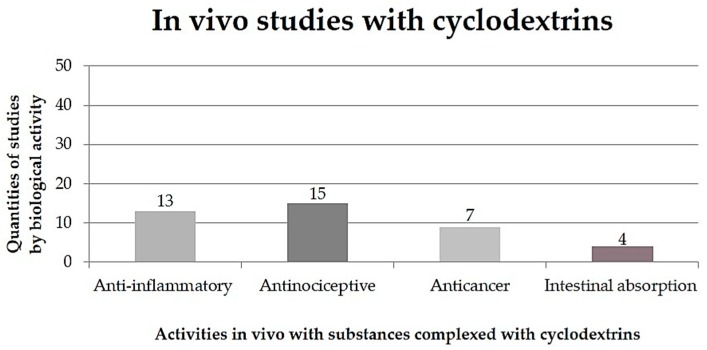
Quantities of in vivo studies performed with substances in an inclusion complex with cyclodextrins, according to works published in the literature in the years 2001 to 2019.

**Figure 3 ijms-20-00642-f003:**
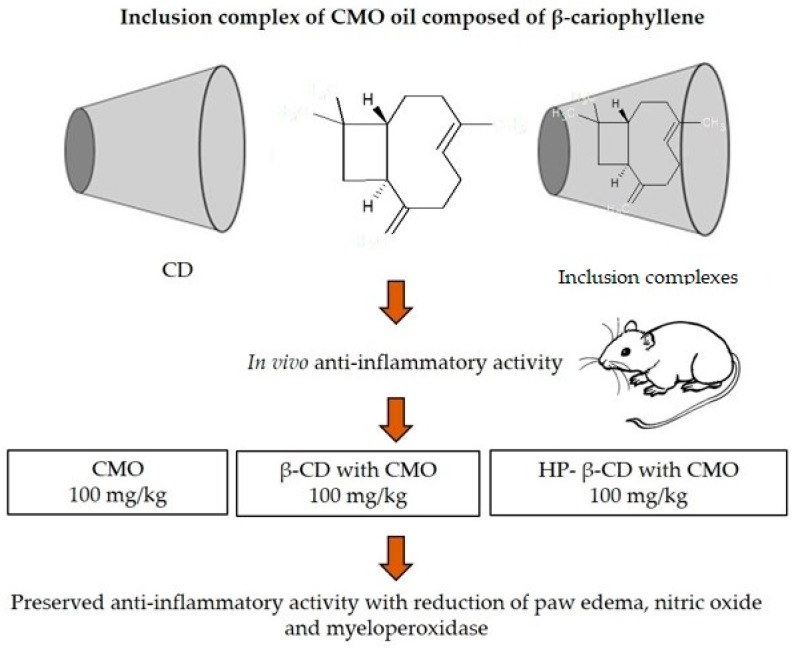
*Copaifera multijuga* oleoresin, inclusion complex and its main major compound β-caryophyllene and anti-inflammatory activity in vivo [32].

**Figure 4 ijms-20-00642-f004:**
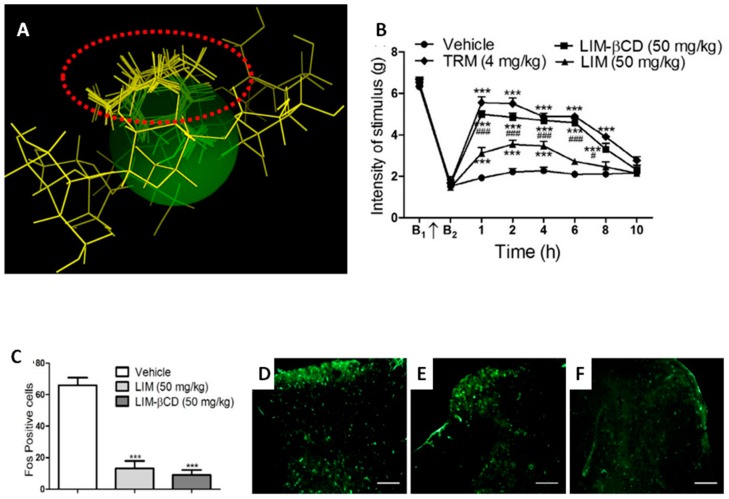
(**A**) Ten possible interactions of limonene (LIM) and CDs obtained through molecular modeling. The green space represents the CD cavity; (**B**) Response time evaluation after administration of vehicle, LIM (50 mg/kg), LIM-βCD (50 mg/kg) or TMR (4 mg/kg) on mechanical hyperalgesia induced by acidic saline in mice. The values are expressed as mean ± SEM. * *p* < 0.05, ** *p* < 0.01 or *** *p* < 0.001 vs. Control group. # *p* < 0.05, ## *p* < 0.01 or ### *p* < 0.001 vs. LIM group (two-way analysis of variance ANOVA followed by Bonferroni test); (**C**) Fos-positive cells in the lumbar spinal cord. Vehicle (**D**), LIM (50 mg/kg) (**E**), LIM-βCD (50 mg/kg) (**F**) were administered 60 min before perfusion. The values are expressed as mean ± SEM (*n* = 6, per group). *** *p* < 0.001 vs. control group (one-way ANOVA followed by Bonferroni test) [42]. Scale bar: 20 μm.

**Figure 5 ijms-20-00642-f005:**
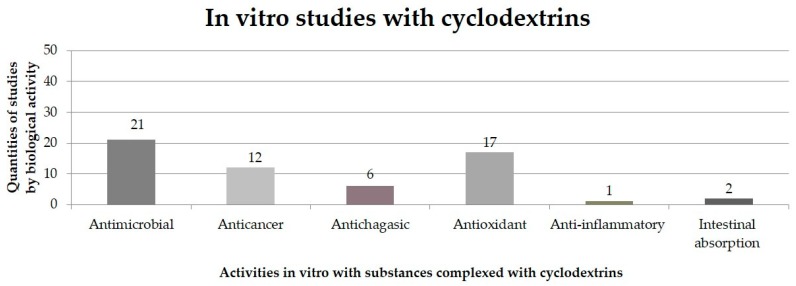
Quantities of in vitro studies performed with substances in inclusion complexes with cyclodextrins, according to works published in the literature from 2001 to 2019.

**Table 1 ijms-20-00642-t001:** In vivo studies of cyclodextrin inclusion complexes.

Biological Activity In Vivo						
CD Type	Complexed Substance	Animal	Dose	Activity	Improved Characteristics	References
β-CD	β-caryophyllene	Mice (S)	10 or 20 mg/kg	Analgesic	All doses tested produced a significant reduction in mechanical hyperalgesia and a significant increase in muscle withdrawal thresholds, without producing any alteration in force	[27]
β-CD	*Ocimum basilicum*	Swiss mice	50 mg/kg	Anti-inflammatory	Inhibited leukocyte recruitment to the peritoneal cavity, and inhibited granuloma formation in mice	[28]
β-CD Hydroxypropyl-β-CD	Oleoresins *Copaifera multijuga*	Swiss mice	100 mg/kg	Anti-inflammatory	This study showed that it is possible to produce inclusion complexes of oleoresins *Copaifera multijuga*, with CDs by the kneading and slurry methods without any change in anti-inflammatory activity	[32]
SBE-β-CD; HP-β-CD	Valdecoxib	Mice (S)	1 mg/kg	Anti-inflammatory	Valdecoxib alone showed slow in vivo absorption giving a maximum % inhibition of edema (16%) after a period of 3 h. Valdecoxib included in the cavity of both the CDs showed high absorption rates in vivo, achieving more than 50% inhibition of edema in the 1h and a maximum percentage of inhibition of edema (66%) after a period of 3 h	[33]
β-CD	Meloxicam	Swiss albino mice (20–25 g)	-	Antinociceptive	A greater percentage inhibition (71.11 ± 1.47%) of the writhing responses was observed with the formulation than with meloxicam alone (45.55 ± 2.05%)	[34]
HP-β-CD	Coumestrol	Wistar rats	10 μM, 50 μM	Anti-inflammatory	The association of 50 μM (66.1%) and 10 μM (56.3%) coumestrol/HP-β-CD induced proliferation and cell migration in inflicted wounds.	[35]
β-CD	(−)-linalool	Swiss mice	40 mg/kg	Antinociceptive	Inclusion complexes with linalool/β-CD revealed that the antinociceptive effect was significantly improved when compared with linalool alone	[36]
β-CD	Essential oil *Lippia grata*	Mice (S)	6, 12, or 24 mg/kg	Antinociceptive	Antinociceptive profile might be linked to the presence of some terpenoids, such as camphor, borneol and b-caryophyllene, and to the activation of the motor cortex, nucleus raphe magnus (NRP) and periaqueductal gray (PAG) cerebral areas involved in pain modulation	[37]
β-CD	LIN	Mice (S)	25 mg/kg (p.o.)	Antihyperalgesic	The inclusion complex prolonged the time-effect, possibly by increasing stability and solubility	[38]
β-CD	Carvacrol	Mice (S)	50 mg/kg (p.o.)	Analgesic	The inclusion complex prolonged the time-effect, possibly through increased stability and solubility	[39]
β-CD	Hecogenin acetate	Swiss mice	20 mg/kg	Antinociceptive	Complexation efficiency of 92%, superior analgesic effect in animal models for orofacial pain at a lower dose when compared to hecogenin alone	[40]
β-CD	Hecogenin acetate	Swiss mice	20 mg/kg	Antihyperalgesic	Oral pretreatment with hecogenin-CD (HA-CD), produced a significant antinociceptive profile and also decreased mechanical hyperalgesia, with HA-CD showing significantly better effects when compared to HA alone	[41]
β-CD	D-limonene	Swiss mice	50 mg/kg	Antihyperalgesic	Longer analgesic duration and reduced Fos protein expression in the dorsal horn of the spinal cord	[42]
β-CD	Curcumin	Mice (S)	100 mg/kg	Anticancer	Curcumin-CDS complexes enhanced curcumin delivery and improved its therapeutic efficacy compared with free curcumin in vivo and in vitro	[43]
β-CD	Farnesol	Swiss mice	50 and 100 mg/kg	Antinociceptive	Improved pharmacological properties when compared to the active compound alone. Biotechnological value in the treatment of some types of dysfunctional pain such as orofacial pain	[44]
β-CD	Essential oil *Cymbopogon winterianus*	Swiss mice	50–200 mg/kg	Antinociceptive	The use of *C. winterianus* essential oil complexed in CD considerably reduced the dose of *C. winterianus* contained in the complex when compared to the doses commonly used for the pure *C. winterianus* described in the literature	[45]
β-CD	LIN	Mice (S)	40 mg/kg	Gastroprotector	The complex revealed that the gastroprotective effect was significantly improved compared with uncomplexed linalool, suggesting that this improvement is related to increased solubility and stability	[46]
SBE-β-CD	Posaconazole	Sprague–Dawley rats (240 ± 20 g)	0.05–4.0 μg/mL	Intestinal absorption	The results demonstrated that the formation of the posaconazole sulfobutylether β-CD inclusion complex significantly improved the bioavailability of posaconazole in comparison with pure posaconazole	[47]
α-CD	Polyurethane graft	Male mice of swiss albino strain with average body weight 20–25 g	500 μg	Anticancer	Efficacy of the sustained release of drug from the graft copolymer without side effects helps suggests a promising novel future drug delivery vehicle for the treatment of melanoma	[48]
SBE-β-CD	Amlodipine	Male New Zealand variety rabbit weighing 1.5–2 kg	-	Anti-inflammatory	Presence of SBE-β-CD in the amlodipine Hydroxypropyl Methylcellulose film improved ocular permeation significantly and could be utilized as a mucoadhesive type formulation for anti-inflammatory activity	[49]
β-CD γ-CD	Limonin	Wistar rats weighing 200–250 g	0.12 mg/kg	Anti-inflammatory	Significant reduction of the volume of the paw edema. Administration of limonin was able to reduce the degree of bone resorption, soft tissue swelling and osteophyte formation, improving articular function in treated animals	[50]
β-CD	Albendazole and ricobendazole	BALB/c mice 8-week-old	30 mg/kg	Anticancer	Solubility was highest when β-CD was used as carrier. This increase in solubility was higher for albendazole, indicating the formation of a more stable complex than with ricobendazole In vivo studies showed that the ABZ:β-CD complex produced a reduction in the tumor growth kinetics on mice with no signs of toxicity	[51]

**Table 2 ijms-20-00642-t002:** In vitro studies of cyclodextrin inclusion complexes.

Biological Activity In Vitro						
CD Type	Complexed Substance	Sample	Concentration	Activity	Improved Characteristics	References
β-CD	*Hyptis martiusii*	*Staphylococcus aureus* *Escherichia coli* *Pseudomonas eruginosa*	1 mg/mL	Antibacterial	Anti-staphylococcal activity of essential oil *H. martiusii* and a synergistic effect when associated with gentamicin against Gram negative bacteria	[26]
SBE-β-CD HP-β-CD	Albendazole	Ovarian tumor cell line OVCAR-3 and human ovarian epithelial cells (HOSE) Ovcar-3 cells	6–8 mg/mL	Anticancer	The addition of 8.0 mg/mL and 7.0 mg/mL of (ABZ) to 40% CD solutions at 25 °C showed maximum complexation with SBE-β-CD & HP-β-CD, respectively, at three days, with 2 weeks stability. [ABZ] complexed with SBE-β-CD showed potent cytotoxicity (in vitro & in vivo) in ovarian tumour cells	[65]
β-CD	Albendazole	4T1 murine mammary carcinoma	0.5 μM	Anticancer	The IC_50_ value obtained for ABZ was 0.56 ± 0.02 μM while for the ABZ:C-β-CD complex was 0.41 ± 0.30 μM. Although ABZ:C-β-CD complex showed lower values of IC_50_ than ABZ, no statistically significant differences were observed	[66]
β-CD	α-bisabolol	*Staphylococcus aureus* *Escherichia coli* *Pseudomonas eruginosa*	1 mg/mL	Antibacterial	Antibacterial effect upon *S. aureus*, in combination with gentamicin	[67]
γ-CD	Alamethicin	*Listeria monocytogenes*	0.0625 mg/mL 1.0 mL	Antibacterial	Images of *L. monocytogenes* exposed to γ-CD/alamethicin complex revealed microbial inactivation	[68]
β-CD	2-nonanone	*Botrytis cinerea*	75.0 mg/L	Antimicrobial	Antimicrobial tests for mycelial growth reduction under atmospheric conditions proved the fungistatic behaviour of the inclusion complexes against *Botrytis cinerea*	[69]
Polyamine-β-CD	Glycyrrhetic acid	Cell lines and normal human lung fibroblast WI-38	-	Anticancer	Satisfactory aqueous solubility, along with high thermal stability of inclusion complexes will be potentially useful for their application in the formulation and design of natural medicines	[70]
α-CD β-CD HP-β-CD Methylated-β-CD	Estragole	DPPH radical scavenging	-	Antioxidant	Antioxidant activity of estragole was increased by the formation of inclusion complexes with CDs. The formation of inclusion complexes allowed a controlled release of estragole	[71]
SBE-β-CD	Nintedanib	EpiIntestinal tissue model	1 μM	Intestinal absorption	The study demonstrated that cyclodextrin complexation increased stability of nintedanib in Phosphate-buffered saline (PBS) (pH 7.4) and simulated intestinal fluid (SIF). Bioactivity of nintedanib also improved. Complexation increased the transport of nintedanib across intestinal membrane and reduced efflux ratio	[72]
HP-β-CD Methylated-β-CD HP-γ-CD	Linalool	*E. coli* and *S. aureus*	-	Antibacterial	A significant amount of linalool was preserved, due to enhancement of the thermal stability of linalool by the cyclodextrin inclusion complexation	[73]
HP-β-CD	Biochanin A (BCA)	*Escherichia coli*, *Pseudomonas aeruginosa*, *Klebsiella pneumoniae*, *Salmonella enteritidis*, *Staphylococcus aureus*, *Candida albicans* and *Aspergillus niger*	0.84–1.69 mg	Antimicrobial	The obtained minimum inhibitory concentration (MIC) values for analyzed bacteria strains were in the range of 0.84–1.69 mg/cm^3^. The prepared inclusion complex expressed less effect against strains *E. coli* and *K. pneumoniae*. Moreover, BCA and inclusion complex did not show activity against fungus *A. niger*. These obtained results indicated that the antimicrobial activity of BCA was not significantly changed after complexation	[74]
β-CD HP-β-CD	Ellagic acid (EA)	*Candida albicans*, *Proteus vulgaris*, *Klebsiella pneumoniae*, *Escherichia coli*, *Pseudomonas aeruginosa*, *Bacillus cereus*, *Bacillus luteus*, and *Listeria monocytogenes*	1 mg	Antimicrobial	The greater antimicrobial activity of inclusion complexes compared with free EA was probably the result of the ability of CDs to release the drug readily from the inclusion complexes. The improved antimicrobial activity could be due to the increased aqueous solubility of EA	[75]
HP-β-CD	Thalidomide	Caco-2 cells	-	Intestinal absorption	Thalidomide was stable in the transport buffer throughout the entire period of 2 h. The acidification of samples prevented the hydrolysis of thalidomide with 93.6 ± 2.2% of the initial drug being detected at the end of the experiment	[76]
β-CD	Ellagic acid (EA)	Protein denaturation and membrane stabilization assay	20 mg/mL	Anti-inflammatory	EACD (20 mg/mL) inclusion complex considerably protected the albumin from denaturation. EACD (20 mg/mL) was able to protect the erythrocyte membrane from lysis induced by heat (32.95%) and hypotonicity (45.72%)	[77]

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
