# Peer review of "Cyclodextrin–Drug Inclusion Complexes: In Vivo and In Vitro Approaches"

_ijms, 2019, doi:10.3390/ijms20030642_

Round 1
Reviewer 1 Report
The review summarizes mostly recent publications dealing with in vitro and in vivo studies of complexes of cyclodextrin derivatives with biologically active compounds and emphasizes the most important findings in each of them. Despite a large number of reviews about the use of cyclodextrins in drug delivery systems, this one could bring another point of view and might be of interest for those working in the field. Nevertheless, the manuscript is in the presented form unacceptable. There are many mostly formal mistakes and omissions which should be corrected.
Some of the errors/problems found (starting with line number if appropriate):
21 .. "cyclodextrins are play" ... -> "cyclodextrins play"
Some relevant most cited review articles about cyclodextrins in drug carrier systems from highly impacted journals should have been mentioned (e.g., Chemical Reviews 1998, Nature Reviews Drug Discovery 2004), and the new findings which appeared since that time emphasized. Also, the recent review about the history of cyclodextrins (Chemical Reviews 2014) should have been mentioned.
39 Abbreviation "CDs" is used without a definition (it should be on line 35).
44 Reference 3, which should explain the formation of inclusion complexes of CD, is an article in Portugal, what is obviously not the best choice (both - language- and journal-wise), considering the existence of a number of highly cited references on the same topic (e.g., Chem. Rev. 1998, J. Inclusion Phenom. Macrocyclic Chemistry 2002).
49 Also, the specific types of CD derivatives could be better referenced to the articles which deal specifically with these types of derivatives and not to other general review articles.
48 ethyl-p-CD -> ethyl-beta-CD (?)
55 The quality of Figure 1 is inferior. The details of structure, labels, and dimensions are almost illegible.
66 CDS -> CDs
72-74 Again, references to the original publications dealing with the pharmacology or toxicology of the specified derivatives should be given (e.g., J. Pharm. Sci. 2010), not only to other general reviews. Reference 16 does not deal with any kidney damage as stated in the sentence preceding it.
82-89 The references are given as a list at the end of the paragraph instead of providing references to each of the six statements/points given in the paragraph.
117 The abbreviation CD-IC is not explained.
148 OBEO is not present in the Abbreviations list
189 What is "kgK1"?
190 What "carboxymethyl"?
192 1h -> 1 h
231 EO abbreviation is not defined
245 CNS abbreviation is not defined in the text
275 "(p b 0.05)" -> "(p < 0.05)" (??)
355 Table 1. The formatting of the table columns does not look very good, and the available space is not efficiently used - wide and almost empty column References, in the first column, full names of CD derivatives are used instead of the previously defined abbreviations.
Consistent spelling should be used throughout the text. The problems - e.g., sulfobutyl x sulphobutyl; β-CD x b-CD x beta-CD x βCD; β-cyclodextrin x βcyclodextrin x b-cyclodextrin; CD-IC x IC CD; HP-β-CD x HPβCD x HPbCD; etc. Overall, other typographic errors are also numerous and must be corrected, e.g., missing or additional spaces (mainly in names of chemicals), an uppercase instead of a lowercase letter and vice versa, errors with commas.
Abbreviations - the list of abbreviations should be sorted alphabetically, and it will also be a good chance to remove duplicates.
Author Response
The review summarizes mostly recent publications dealing with in vitro and in vivo studies of complexes of cyclodextrin derivatives with biologically active compounds and emphasizes the most important findings in each of them. Despite a large number of reviews about the use of cyclodextrins in drug delivery systems, this one could bring another point of view and might be of interest for those working in the field. Nevertheless, the manuscript is in the presented form unacceptable. There are many mostly formal mistakes and omissions which should be corrected.
Some of the errors/problems found (starting with line number if appropriate):
21 .. "cyclodextrins are play" ... -> "cyclodextrins play"
Answer: We corrected this. Thanks.
Some relevant most cited review articles about cyclodextrins in drug carrier systems from highly impacted journals should have been mentioned (e.g., Chemical Reviews 1998, Nature Reviews Drug Discovery 2004), and the new findings which appeared since that time emphasized. Also, the recent review about the history of cyclodextrins (Chemical Reviews 2014) should have been mentioned.
Answer: We thank you for the suggested articles for this review. We have mentioned all the excellent articles suggested.
39 Abbreviation "CDs" is used without a definition (it should be on line 35).
Answer: We corrected this. Thanks.
44 Reference 3, which should explain the formation of inclusion complexes of CD, is an article in Portugal, what is obviously not the best choice (both - language- and journal-wise), considering the existence of a number of highly cited references on the same topic (e.g., Chem. Rev. 1998, J. Inclusion Phenom. Macrocyclic Chemistry 2002).
Answer: We enriched the references in relation to this topic.
49 Also, the specific types of CD derivatives could be better referenced to the articles which deal specifically with these types of derivatives and not to other general review articles.
Answer: We improved this part with more specific references.
48 ethyl-p-CD -> ethyl-beta-CD (?)
Answer: We corrected this. Thanks.
55 The quality of Figure 1 is inferior. The details of structure, labels, and dimensions are almost illegible.
Answer: We corrected this. Thanks.
66 CDS -> CDs
Answer: We corrected this. Thanks.
72-74 Again, references to the original publications dealing with the pharmacology or toxicology of the specified derivatives should be given (e.g., J. Pharm. Sci. 2010), not only to other general reviews. Reference 16 does not deal with any kidney damage as stated in the sentence preceding it.
Answer: We have improved the references for this section and removed the incorrect reference.
82-89 The references are given as a list at the end of the paragraph instead of providing references to each of the six statements/points given in the paragraph.
Answer: We have amended this and provided references to each of the six statements.
117 The abbreviation CD-IC is not explained.
Answer: We corrected this. Thanks.
148 OBEO is not present in the Abbreviations list
Answer: We corrected this. Thanks.
189 What is "kgK1"?
Answer: We corrected this. Thanks.
190 What "carboxymethyl"?
Answer: We corrected this. Thanks.
192 1h -> 1 h
Answer: We corrected this. Thanks.
231 EO abbreviation is not defined
Answer: We corrected this. Thanks.
245 CNS abbreviation is not defined in the text
Answer: We corrected this. Thanks.
275 "(p b 0.05)" -> "(p < 0.05)" (??)
Answer: We corrected this. Thanks.
355 Table 1. The formatting of the table columns does not look very good, and the available space is not efficiently used - wide and almost empty column References, in the first column, full names of CD derivatives are used instead of the previously defined abbreviations.
Answer: We corrected this. Thanks. The references are each cited in the last column. In the tables the references are highlighted by work explored according to the biological activity performed with inclusion complex.
Consistent spelling should be used throughout the text. The problems - e.g., sulfobutyl x sulphobutyl; β-CD x b-CD x beta-CD x βCD; β-cyclodextrin x βcyclodextrin x b-cyclodextrin; CD-IC x IC CD; HP-β-CD x HPβCD x HPbCD; etc. Overall, other typographic errors are also numerous and must be corrected, e.g., missing or additional spaces (mainly in names of chemicals), an uppercase instead of a lowercase letter and vice versa, errors with commas.
Answer: We corrected this. Thanks.
Abbreviations - the list of abbreviations should be sorted alphabetically, and it will also be a good chance to remove duplicates.
Answer: We corrected this. Thanks.
Reviewer 2 Report
Int.J.Mol.Sci_2018_401435
Review
The manuscript untitled “Review of cyclodextrin-drug inclusion complexes: in vivo and in vitro approaches” by Simone Braga Carneiro, Fernanda Ílary Costa Duarte, Luana Heimfarth, Jullyana de Souza Siqueira Quintans, Lucindo José Quintans-Júnior, Valdir Florêncio da Veiga Júnior, Ádley Antonini Neves de Lima
The manuscript represents the review devoted to analysis of cyclodextrin-drug inclusion complexes, to theirs in vitro and in vivo evaluations. This area of organic chemistry is quite deeply researched and these materials are important due to the biotechnological innovative potential of cyclodextrin complexes. Thus, the present manuscript to be seems timely.
These materials can be useful to chemists, biochemists and biologists.
The manuscript not calls any objections and could be published in Int.J.Mol.Sci in the present form with revision.
Several details should be noted.
1. From the title should remove the word "review".
2. Line 105. Section 2. “In vivo…” Number 2 should be added.
3. Figure 2. In the caption for Fig. 2 a period of time for published works should be added.
4. Figure 5. The same.
5. Figure 2 shows 9 literary references to work on anticancer activity. In the text of the section 2.3. "Anticancer activities of cyclodextrin inclusion complexes" mentions only two articles with in vivo studies and one with in vitro study. What is the reason?
6. In the caption under Fig. 3 should be given the abbreviation of CMO.
7. Figure 6. Why is this particular figure 6 given? What's the point?
8. Line 330. “In order to reduce bioavailability…” There is no error?
9. Line 331. What water-soluble cyclodextrin was investigated? Ref. [47].
10. Line 400. “…thus CDs haves been used successfully used…”
11. Table 1 should be mentioned in the text. The data in Table 1 are not ordered.
12. Why section 3 does not discuss the in vitro data on anti-inflammatory activity and intestinal absorption mentioned in Fig. 5?
13. References are presented casually. You should carefully check the references.
[63] The title of the article “Cyclodextrins and Antioxidants” is repeated.
[70] J. Cancer, pp.?
[86] Volume? Pages?
[92] Line 456, ??
14. Abbreviations must be in alphabetical order.
Lines 96. CDs is the designation of cyclodextrin or cyclodextrins?
Lines 97 and 103, a repeat.
Line 113, superfluous designation, since both β-cyclodextrin and kamebakaurin KA are already indicated.
Line 125. The same.
Line 128. The same.
Line 118. CVD => Differential Scanning Calorimetry. It's not a mistake?
15. Line 337. b => β.
16. Read the entire text carefully. Many minor fixes should be made. For example, remove the intervals from the slash (line 510, 7.0 mg / ml, must be 7.0 mg/ml) and the like.
Author Response
The manuscript represents the review devoted to analysis of cyclodextrin-drug inclusion complexes, to theirs in vitro and in vivo evaluations. This area of organic chemistry is quite deeply researched and these materials are important due to the biotechnological innovative potential of cyclodextrin complexes. Thus, the present manuscript to be seems timely.
These materials can be useful to chemists, biochemists and biologists.
The manuscript not calls any objections and could be published in Int.J.Mol.Sci in the present form with revision.
Several details should be noted.
1. From the title should remove the word "review".
Answer: We corrected this. Thanks.
2. Line 105. Section 2. “In vivo…” Number 2 should be added.
Answer: We corrected this. Thanks.
3. Figure 2. In the caption for Fig. 2 a period of time for published works should be added.
Answer: We corrected this. Thanks.
4. Figure 5. The same.
Answer: We corrected this. Thanks.
5. Figure 2 shows 9 literary references to work on anticancer activity. In the text of the section 2.3. "Anticancer activities of cyclodextrin inclusion complexes" mentions only two articles with in vivo studies and one with in vitro study. What is the reason?
Answer: We added about more of anti-cancer studies highlighted in Figure 2 to item 2.3 and table 1.
6. In the caption under Fig. 3 should be given the abbreviation of CMO.
Answer: We corrected this. Thanks.
7. Figure 6. Why is this particular figure 6 given? What's the point?
Answer: We removed this figure as the study and the results are already well described in the text.
8. Line 330. “In order to reduce bioavailability…” There is no error?
Answer: We corrected this. Thanks.
9. Line 331. What water-soluble cyclodextrin was investigated? Ref. [47].
Answer: The cyclodextrin used is γ-cyclodextrin. We added this information to the text as requested.
10. Line 400. “…thus CDs haves been used successfully used…”
Answer: We corrected this. Thanks.
11. Table 1 should be mentioned in the text. The data in Table 1 are not ordered.
Answer: We corrected this. Thanks.
12. Why section 3 does not discuss the in vitro data on anti-inflammatory activity and intestinal absorption mentioned in Fig. 5?
Answer: Because there are few in vitro studies for these activities, we have decided to add these studies to the table 2 which shows various data on in vitro studies with inclusion complexes.
13. References are presented casually. You should carefully check the references.
[63] The title of the article “Cyclodextrins and Antioxidants” is repeated.
[70] J. Cancer, pp.?
[86] Volume? Pages?
[92] Line 456, ??
Answer: We have made all the requested corrections. Thank you for your careful review of our work.
14. Abbreviations must be in alphabetical order.
Lines 96. CDs is the designation of cyclodextrin or cyclodextrins?
Lines 97 and 103, a repeat.
Line 113, superfluous designation, since both β-cyclodextrin and kamebakaurin KA are already indicated.
Line 125. The same.
Line 128. The same.
Line 118. CVD => Differential Scanning Calorimetry. It's not a mistake?
Answer: We have made all the requested corrections.
15. Line 337. b => β.
Answer: We corrected this. Thanks.
16. Read the entire text carefully. Many minor fixes should be made. For example, remove the intervals from the slash (line 510, 7.0 mg / ml, must be 7.0 mg/ml) and the like.
Answer: We corrected this. Thanks.
Round 2
Reviewer 1 Report
Figure 1 got more legible, indeed; unfortunately, most of the monosaccharide units of CDs do not look like glucose anymore (maybe allose?). I would recommend to use "ChemDraw - templates - supramolecular" to get more correct structures of all three CDs.
Author Response
Answer:
Cyclodextrins were drawn in Chemdraw and the resolution of figure 1 was improved.
